

# Transcriptome analysis during 4-vinylcyclohexene diepoxide exposure-induced premature ovarian insufficiency in mice

Yi Li[1], Ruifen He[1], Xue Qin[1], Qinying Zhu[1], Liangjian Ma[1] and Xiaolei Liang[2]

[1] The First Clinical Medical College, Lanzhou University, Lanzhou, Gansu, China
[2] Gansu Provincial Clinical Research Center for Gynecological Oncology, the First Hospital of Lanzhou University, Lanzhou, Gansu, China

## ABSTRACT

The occupational chemical 4-Vinylcyclohexene diepoxide (VCD) is a reproductively toxic environmental pollutant that causes follicular failure, leading to premature ovarian insufficiency (POI), which significantly impacts a woman's physical health and fertility. Investigating VCD's pathogenic mechanisms can offer insights for the prevention of ovarian impairment and the treatment of POI. This study established a mouse model of POI through intraperitoneal injection of VCD into female C57BL/6 mice for 15 days. The results were then compared with those of the control group, including a comparison of phenotypic characteristics and transcriptome differences, at two time points: day 15 and day 30. Through a comprehensive analysis of differentially expressed genes (DEGs), key genes were identified and validated some using RT-PCR. The results revealed significant impacts on sex hormone levels, follicle number, and the estrous cycle in VCD-induced POI mice on both day 15 and day 30. The DEGs and enrichment results obtained on day 15 were not as significant as those obtained on day 30. The results of this study provide a preliminary indication that steroid hormone synthesis, DNA damage repair, and impaired oocyte mitosis are pivotal in VCD-mediated ovarian dysfunction. This dysfunction may have been caused by VCD damage to the primordial follicular pool, impairing follicular development and aggravating ovarian damage over time, making it gradually difficult for the ovaries to perform their normal functions.

## INTRODUCTION

In today's society, people are exposed to a wide range of synthetic chemicals (*Priya et al., 2021*), some of which can mimic hormones, disrupting natural hormonal processes and thus interfering with regulatory and reproductive mechanisms (*Priya et al., 2021*). Occupational chemical 4-Vinylcyclohexene diepoxide (VCD) is a processed derivative of 4-vinylcyclohexene (VCH) (*Huff, 2001*). VCD serves as a reactive diluent and chemical intermediate in epoxy and dicyclohexyl resin, and is separated during rubber tire,

Corresponding author
Xiaolei Liang, liangxl07@lzu.edu.cn

pesticide, plasticizer, and flame retardant production processes and released directly into the environment (*Rappaport & Fraser, 1977*). According to the guidelines provided by the National Institute for Occupational Safety and Health (NIOSH) and the American Conference of Governmental Industrial Hygienists (ACGIH), the permissible exposure level for VCD is between 10–0.1 ppm, averaged over an 8–10-hour work period. However, the risk of overexposure persists, particularly through direct skin and oral contact, even when airborne levels fall below the exposure limit. Research on the reproductive system has shown that VCD can cause atrophy of the ovaries and uterus in mice (*National Toxicology Program, 1986*; *National Toxicology Program, 1989*). Therefore, it is important to study VCD exposure in the population and investigate its impact on female reproductive ability.

The number of follicles in humans is limited, and long-term exposure to toxic substances can lead to the depletion of primordial follicles, which can lead to infertility (*Karwacka et al., 2019*). Research has shown that exposure to VCD triggers the depletion of primordial follicular reserves in the ovaries of young female rodents (*Mayer et al., 2002*). The toxic effects of VCD can lead to premature ovarian insufficiency (POI) through the expelling of primordial follicles from the ovaries (*Mayer et al., 2004*). POI is typically characterized by complete follicular failure or ovarian dysfunction of premenopausal ovaries and is one of the most common causes of infertility in women under 40 years old (*Nelson, 2009*; *Yeganeh et al., 2019*). Several previous studies have shown that sustained exposure to VCD triggers apoptosis through the activation of caspase-mediated apoptotic pathways (*Hu et al., 2001*; *Takai et al., 2003*) and the KIT/KITL signaling pathway (*Carlsson et al., 2006*; *Fernandez et al., 2008*), leading to follicle loss (*Kappeler & Hoyer, 2012*; *Carolino et al., 2019*). However, the specific molecular mechanism by which VCD causes POI is unclear and requires further research.

This study successfully established a POI mouse model using VCD. Changes in hormone levels and the number of follicles were compared in normal mice and POI mice after 15 and 30 days of VCD treatment. Transcriptomics was used to evaluate the differences in expression among the three groups and to explore the molecular mechanism of POI caused by VCD. These findings help reveal the pathogenesis of POI caused by VCD exposure and broaden the direction of related research; these findings may also lead to more targeted treatment strategies for POI patients.

## MATERIAL AND METHODS

### Chemical

4-Vinylcyclohexene diepoxide (V820469-5 ml) and sesame oil (S905724-50 ml) were purchased from Macklin, Inc. (Macklin, Shanghai, China). VCD was mixed with sesame oil and administered to mice *via* intraperitoneal injection.

### Feeding conditions for experimental animals

A total of 30 specific pathogen free (SPF) seven-week-old female C57BL/6 mice were purchased from the Jiangsu Huachuang Sino Pharmaceutical Technology Co., Ltd. (Jiangsu, China). The mice were acclimatized for one week before the formal experiment, kept in separate cages (six cages, with five mice per cage), and had free access to food and water.

The environment provided a normal circadian rhythm and a constant temperature and humidity (24 $\pm$ 2 °C/40%). Ethics committee permission was obtained from Lanzhou University's First Hospital for all animal experiments (LDYYLL2023-105).

## Animal model establishment and sample collection

Complete randomization was used to divide the mice into three groups: a control group ($n = 10$), a V15 experimental group ($n = 10$), and a V30 experimental group ($n = 10$). The control group was injected with sesame oil, and the two experimental groups were given VCD dissolved in sesame oil at a concentration of 160 mg/kg *via* intraperitoneal injection. The injection required for 15 consecutive days. The dose of VCD were based on published reports (*Cao et al., 2020*) and our pre-experiments. Simultaneous vaginal cytology cell smears were made during the 15-day modeling period for observation of the estrous cycle.

After modeling, the mice were fed normally for different time durations. For the V15 group, serum and ovaries were harvested on day 15 after injection. For the CON and V30 groups, serum and ovaries were collected on day 30. All mice were euthanized *via* transperitoneal injection of pentobarbital in accordance with the American Veterinary Medical Association (AVMA) Guidelines for Euthanasia of Animals before serum collection and organ (ovary) harvesting. After euthanasia, the eyes of the mice were immediately removed to collect blood, and the blood was allowed to sit overnight in a 4 °C refrigerator. The serum was collected by centrifugation at 3,000 rpm for 10 min. The left ovaries of the mice were stored in a −80 °C freezer, and the right ovaries were completely immersed in tissue fixative for 24 h before being used for subsequent experiments.

## Hormone measurement

Serum samples from 10 mice in each group (a total of 30 mice) were used to measure hormone levels. The serum anti-mullerian hormone (AMH), follicle stimulating hormone (FSH) and estrogen ($E_2$) levels of the mice were determined using matching enzyme-linked immunosorbent assay (ELISA) kits purchased from the Jingmei Biotech Company (Jiangsu, China). The standard curve range for AMH was 0–180pg/ml, and the lower limit of quantification was 11.25 pg/ml; the standard curve range for FSH was 16 U/L, and the lower limit of quantification was 1U/L; the standard curve range for $E_2$ was 0–200 pmol/L, and the lower limit of quantification was 12.5 pmol/L. The test was performed in strict accordance with the kit instructions.

## Follicle count

The right side of the mouse ovary was sectioned ($n = 10$ from each group). Ovaries were immersed in paraformaldehyde and fixed by paraffin embedding. The embedded ovaries were serially sectioned (5 $\mu$m thick sections), and the sections were stained with hematoxylin-eosin (HE). Observation of ovarian morphology was carried out under an orthoptic microscope (OLYMPUS, BX51; Olympus, Tokyo, Japan). The number of follicles at each stage was counted in one of every five sections, and only follicles with visible oocyte nuclei, including primordial follicles, primary follicles (PFs), secondary follicles (SFs), antral follicles (AFs) and the corpus luteum (CLs), were recorded. Ovarian tissue sections

in this article were photographed using E3ISPM06300KPA (Hangzhou ToupTek Photonics Co., Ltd., Hangzhou, China).

## RNA-seq

Ovarian tissue RNA was extracted using a TRIzol kit (Invitrogen; $n = 5$ from each group). The RNA Nano 6000 Assay Kit on the Bioanalyzer 2100 system (Agilent Technologies, Santa Clara, CA, USA) was used to evaluate RNA integrity. Poly(A)-tailed mRNA was enriched using oligo (dT) magnetic beads. The obtained mRNA was then fragmented using a fragmentation buffer with divalent cations. Fragmented mRNA served as a template for the synthesis of the first cDNA strand using random oligonucleotides as primers in the presence of M-MuLV reverse transcriptase. Following RNaseH-mediated RNA degradation, the second cDNA strand was synthesized using DNA polymerase I and dNTPs. Purified double-stranded cDNA was subjected to end repair, A-tailing, and adapter ligation, followed by size selection using AMPure XP beads targeting cDNA fragments ranging from 370 to 420 bp. PCR amplification was performed on the selected fragments, followed by purification using AMPure XP beads to obtain the final library. After library construction, preliminary quantification was conducted using a Qubit 2.0 Fluorometer, and libraries were diluted to 1.5 ng/μl. The insert size of the libraries was assessed using an Agilent 2100 bioanalyzer. Libraries underwent quantitative real-time PCR (qRT-PCR) for accurate quantification of effective library concentration (library concentration > 1.5 nM) to ensure library quality. Then, the cDNA libraries were sequenced using the Illumina Nova Seq platform. The obtained raw data were filtered to eliminate reads with adapters, poly-N (which denotes undetermined base information), or low-quality reads (reads with Qphred $\leq 5$ bases that accounted for more than 50% of the entire read length). HISAT2 v2.0.5 was used to compare the reads with the reference genome. The differential expression analysis was conducted using DESeq2 software (version 1.20.0). The Benjamin-Hochberg procedure (B-H FDR) was used with a screening criterion of $| \log2 (\text{Fold Change}) | \geq 1$ and $p$-adjusted ($p$adj) $\leq 0.05$ was used to detect DEGs between the treatment and control groups. The statistical power of these experimental data was as follows: RNASeqPower of CON $vs$ V15 is 0.98984, and CON $vs$ V30 is 0.99862. Correlation coefficients within and between group samples were also calculated based on FPKM values for all genes in each sample and produced as heatmaps (Fig. S1).

## GO and KEGG analyses of DEGs

The ClusterProfiler software was used to analyze the differential gene sets for GO function enrichment and KEGG pathway enrichment. Gene Ontology (GO) enrichment analysis was used to determine the protein functions of the genes, including biological process (BP), cellular component (CC) and molecular function (MF), with $p$adj $\leq 0.05$ as the threshold of significance for enrichment. Kyoto Encyclopedia of Genes and Genomes (KEGG) analysis was used to determine the gene function, genomic information, and functional information of the DEGs, with $p$adj $\leq 0.05$ serving as the threshold for significant enrichment.

## PPI network construction

The STRING protein interaction database (http://string-db.org) was used to analyze the protein–protein interaction (PPI) networks, which were then visually edited with Cytoscape software (version 3.10) and analyzed *via* the degree method to identify hub nodes in the networks. Hub genes may include genes that play relatively important roles in biological processes.

## RT-PCR

RT-PCR was used to analyze several key genes screened in the RNA-seq results ($n = 5$ from each group). Total RNA was extracted using the TRIzol method (Invitrogen, USA) and all steps were performed at room temperature (20 °C–25 °C). The RNA was reverse transcribed to obtain cDNA using the PrimeScript RT Kit and the genomic DNA scavenger (Takara, Japan), and the experimental procedure was carried out in full accordance with the instructions of the kit. The cDNA was extracted using the SYBR Green PCR kit (Takara, Japan) on a StepOnePlus Real-Time PCR system (Applied Biosystems, Waltham, MA, USA) for RT-PCR. Table 1 lists the primer sequences that were designed.

## Statistical analysis

All the statistical analyses were performed using GraphPad Prism 8 (GraphPad Software, La Jolla, CA, USA). One-way ANOVA was used to determine significant differences among the three groups. $P$-values less than 0.05 were considered to indicate statistically significant. All the data were expressed as the mean ±SD. Experimental manipulation and data analysis were carried out by two separates people.

## RESULTS

### VCD affects sex hormone secretion and follicular growth in mice

Previous research has shown that the difference in the ovaries of VCD-model mice occurs 30 days after the end of drug administration (*Cao et al., 2020*). The other commonly used mouse models of POI, such as cyclophosphamide (CTX), usually show difference in the ovaries on the 15th day (*Dai et al., 2023*). Based on these previous results, the present study used the 15th and 30th days as the observation time points and collected the ovaries and serum at those time points for subsequent experiments (Fig. 1A). The results showed AMH and $E_2$ levels were significantly lower in both the V15 and V30 group than in the CON (Fig. 1B). FSH tended to increase in the experimental groups, but the difference was not statistically significant (Fig. 1B). Next, the numbers of primordial follicles, primary follicles (PFs), secondary follicles (SFs), antral follicles (AFs), and corpora luteum (CL) was counted in the three groups (Figs. 1C and 1D). The results showed the number of follicles of all five species was significantly reduced in both V15 and V30. During the modeling period, the estrous cycles of each group of mice were monitored, revealing that the estrous cycle rhythm was disrupted in the mice injected with VCD (Fig. S2 and Table S1). These results are consistent with the clinical features of POI. Based on the differences in the apparent morphological characteristics of the mice in the two experimental groups, these results suggest that the damage caused by VCD to the ovaries of the mice intensified over time.

**Table 1  Specific primer sequences.**

| Gene | Primer sequence (5′–3′) |
|------|------------------------|
| BUB1 | AGAATGCTCTGTCAGCTCATCT |
|      | TGTCTTCACTAACCCACTGCT |
| BUB1B | GGCTGAAGAATACGAAGCTAGAG |
|       | AGCCTTGCGTTCAATCCCTT |
| CDC45 | GAGGTTCCTGCCTACGACG |
|       | TCCTGTTTCGCTCCACTATCT |
| BRCA2 | CCCCACGGTTATGAACCACAG |
|       | TAGCAACATCTACCACAGGGT |
| FANCA | CAGTGTAAAAGGTTGCGTCTCA |
|       | GCCGGGCAATCATTAGTAAGGAA |
| CYP11A1 | CCAAGGATGCGTCGATACTC |
|         | CGGTCTTTCTTCCAGGCATC |
| CYP17A1 | ATTCCACGAAGTGTACTGTGC |
|         | AGGGCTTGCTCATAACCACG |
| HSD17B1 | ATTCCACGAAGTGTACTGTGC |
|         | AGGGCTTGCTCATAACCACG |
| HSD3B6 | TGGACAAAGTATTCCGACCAGA |
|        | GGCACACTTGCTTGAACACAG |
| FSHR | TGCTCTAACAGGGTCTTCCTC |
|      | TCTCAGTTCAATGGCGTTCCG |

## Transcriptomics-based identification of DEGs

A transcriptome analysis was performed to reveal the differences in transcript profiles between the control group and experimental groups. A principal component analysis (PCA) revealed notable differences between samples, with tightly clustered samples from each group being relatively separated from one another, indicating significant distinctions in gene expression profiles between the groups (Fig. 2A). The control group was used as a reference point to identify disparities in gene expression. Then, a heatmap (Fig. 2B) and volcano plots (Figs. 2C and 2D) were generated to visualize the differences between the groups. The results showed that nine genes were upregulated and 458 genes were downregulated in the V15 group in contrast to the control group (Fig. 2C). Similarly, 488 genes were upregulated and 1,861 genes were downregulated in the V30 group (Fig. 2D). This finding suggests that VCD affected gene expression, with a significant difference in the expressed gene profiles of the transcriptomes observed between the groups. The complete results of the difference analysis are shown in Tables S2 and S3.

## Enrichment analysis of DEGs

Enrichment analyses of the differentially expressed genes were then performed. The top 10 significant enrichment results of the GO terms and the KEGG pathways in the V15 and V30 groups are shown in Figs. 3 and 4 (if there were fewer than 10 significant results, the first 10 results are displayed). In group V15, too few genes were upregulated, so no significant enrichment results were obtained (Figs. 3A and 3C). The GO and KEGG pathway

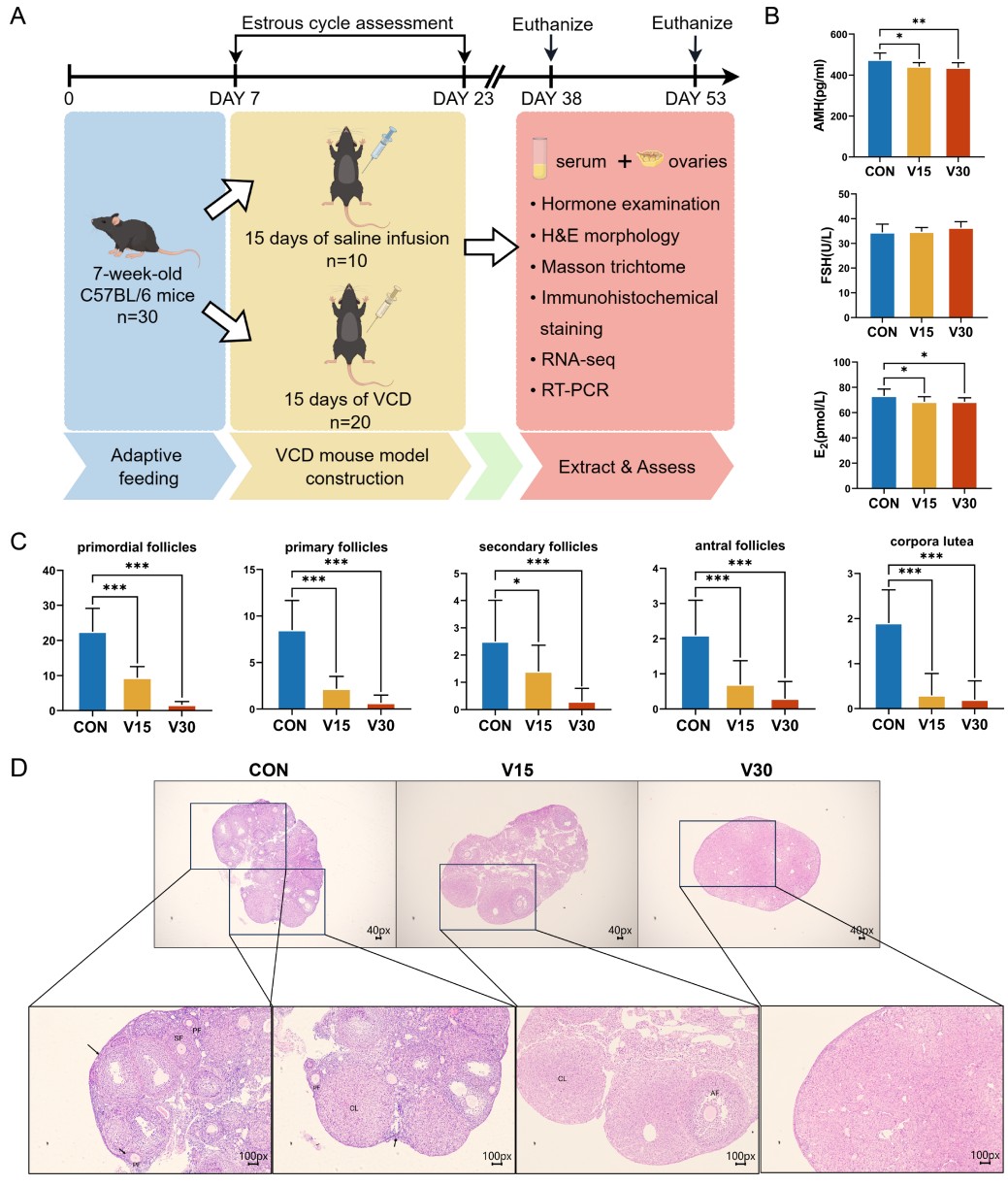

**Figure 1** **Establishment of mouse POI models.** (A) Schematic diagram of the experimental flow. The two execution time points for mice were: day 15 and day 30. (B) Serum AMH, FSH, and $E_2$ levels were measured using ELISA. (C) Follicle counts of control mice and experimental mice at each stage on 15th day and 30th day after administration of the drug. (D) Representative ovary sections of control mice and experimental mice. Black arrows indicate primordial follicles; PF, primary follicle; SF, secondary follicle; AT, atretic follicles; CL, corpora lutea. The data are presented as the mean ± SD ($n = 10$ per group). Notes: $*p < 0.05$ compared to the control group; $**p < 0.01$ compared to control group; $***p < 0.001$ compared to control group. Mouse figure source credit: By Figdraw.

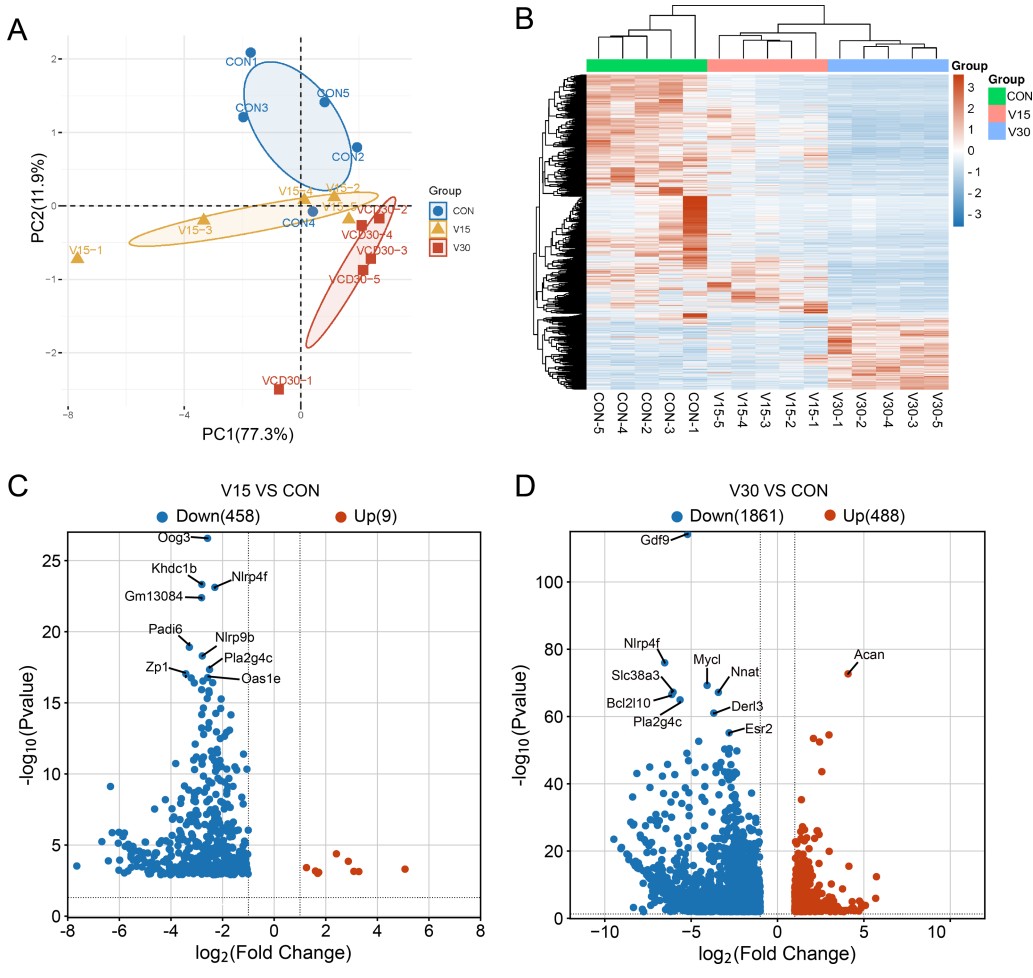

**Figure 2** **Comparison of mRNA expression differences in three groups.** (A) PCA of gene expression values (FPKM) for the three sets of samples. (B) Clustering heatmap of DEGs. The expression of DEGs from high to low is indicated by the change from red to blue. (C) The differential gene volcano map between groups V15 and CON. (D) The differential gene volcano map between groups V30 and CON.

enrichment analyses results showed the downregulated DEGs were related mainly to the motor protein family and mitosis-related molecular activities, such as cilium movement and axoneme assembly (Figs. 3B and 3D). Richer results were obtained from the GO and KEGG pathway analyses for group V30: the downregulated DEGs were associated with the cell cycle; DNA damage repair-related pathways such as Fanconi anemia; homologous recombination, and steroid anabolism, including meiosis, ovarian steroidogenesis, and oocyte homologous recombination (Figs. 4B and 4D). The upregulated DEGs were significantly associated with steroid biosynthesis (Figs. 4A and 4C). There were common pathways between the two groups, but they also had unique differences; more results were obtained at V30, which may indicate that the effects of VCD on the mouse ovary increase over time.

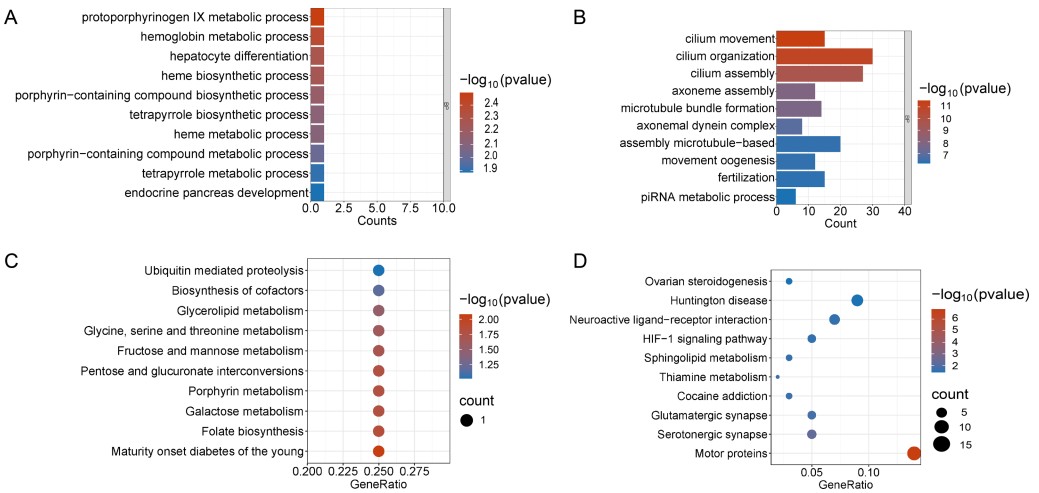

**Figure 3** **Functional enrichment analyzes including GO and KEGG.** (A and B) The top 10 significantly enriched BP terms from the GO analyses of upregulated and downregulated DEGs of V15 *VS* CON. (C and D) The top 10 significantly enriched KEGG pathways of upregulated and downregulated DEGs among the CON group and the V15 group. If there are less than 10 significant results, the top 10 results will be displayed. BP, biological process; GO, Gene Ontology; KEGG, Kyoto Encyclopedia of Genes and Genomes.

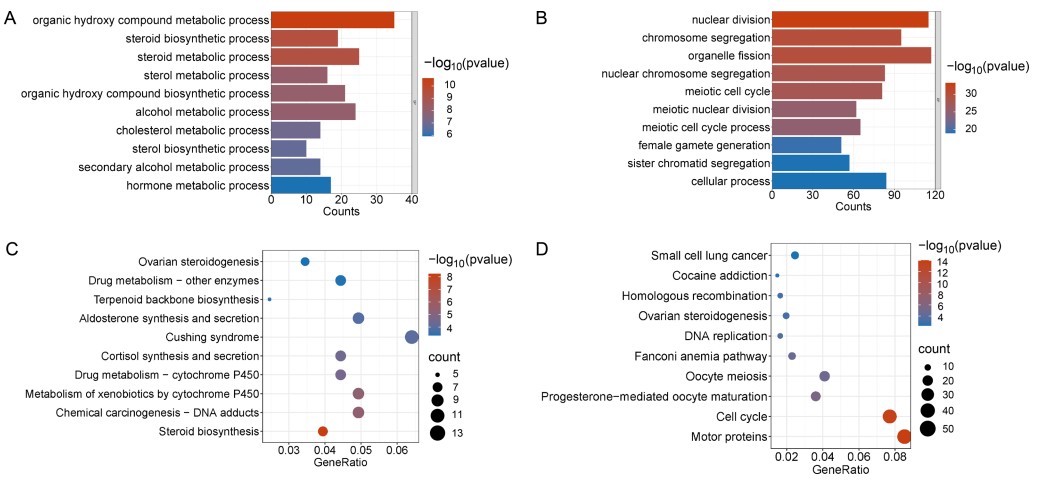

**Figure 4** **Functional enrichment analyzes including GO and KEGG.** (A and B) The top 10 significantly enriched BP terms from the GO analyses of upregulated and downregulated DEGs of V30 *VS* CON. (C and D) The top 10 significantly enriched KEGG pathways of upregulated and downregulated DEGs among the CON group and the V30 group. If there are less than 10 significant results, the top 10 results will be displayed. DEGs, differentially expressed genes; BP, biological process; GO, Gene Ontology; KEGG, Kyoto Encyclopedia of Genes and Genomes.

## PPI Network Analyses of DEGs

The differential gene protein interaction networks were analyzed using the interactions in the STRING database and two PPI networks based on the differential genes of V15 *vs* CON and V30 *vs* CON were produced using Cytoscape_v3.10.0. The hub nodes were identified

from this PPI network based on node degree, and the results were arranged in descending order of node degree (Table S6). For V15 *vs* CON, the top three key genes were Oog3, Oog4, Oog2 (Fig. 5A). And according to the analysis results, C87414, Pramef1, C87499, Gm10436, Gm13084, Gm13023, Gm13103, Oog1, C87977, Pramef20, C87414-2, Gm3106, and D5Ertd577e share the same node degree. Oog3, Oog4, Oog2 and Oog1 belong to a family of genes, all of which have been found to be located on chromosome 4 and are expressed throughout oogenesis and preimplantation embryonic development (*Dadé et al., 2003*). No studies related to female reproduction are available for the remaining genes. For V30 *vs* CON, the five genes with the most commonalities were KIF11, BUB1B, CDC20, BUB1, and CENPE (Fig. 5B), which all play important roles in meiosis in oocytes (*Duesbery et al., 1997*; *Chen et al., 2010*; *Liu et al., 2010*; *Kovacovicova et al., 2016*). This PPI network construction and analysis provides insights into the possible causative agent of VCD.

### Expression levels of target genes in the ovaries

To validate the RNA sequencing results and explore the possible genes and pathways related to altered ovarian function in mice during VCD modeling, 10 key genes were selected: BUB1, BUB1B, CDC45, CYP11A1, CYP17A1, HSD17B1, HSD3B6, FSHR, BRCA2, and FANCA. These genes differed significantly between the groups according to the results of the difference analysis (Tables S2 and S3). Based on the GO and KEGG results, these 10 genes were clustered in pathways with strong physiological relevance to POI, such as the cell cycle, DNA damage and steroid anabolism. The results of quantitative real-time PCR were consistent with the results of the RNA-seq analysis. BUB1, BUB1B, and CDC45 were mapped to the oocyte meiotic cycle pathway. As shown in Figs. 6A, 6B and 6C, the expression levels of these genes showed a downward trend and they were statistically significant in the V30 group. BRCA2 and FANCA are key genes of the Fanconi anemia pathway, and their expression was also significantly lower in V30 than in the control mice (Figs. 6D and 6E). Notably, the expression trends of key genes involved in ovarian steroid hormone synthesis were different between V15 and V30 mice, with the expression levels of CYP17A1 and HSD3B6 increasing in the V30 group (Figs. 6F and 6G), and the expression level of HSD17B1 gradually decreasing, reaching significance at V15 (Fig. 6H). CYP11A1 and FSHR did not significantly differ between the groups (Figs. 6I and 6J). These findings underscore the dynamic alterations in gene expression patterns associated with VCD-induced ovarian dysfunction, shedding light on potential molecular mechanisms underlying the pathogenesis of POI.

## DISCUSSION

Several previous studies have suggested that exposure to VCD can lead to atretic degeneration of primordial and primary follicles in female rodents and primates (*Devine et al., 2002*; *Appt et al., 2006*). Chemical destruction of the follicular pool in women is of concern because it can lead to POI, but the pathogenesis of POI is poorly understood. This study established a mouse model of VCD-induced POI and performed transcriptome analysis of mouse ovaries in an attempt to investigate how VCD affects ovarian function.
A

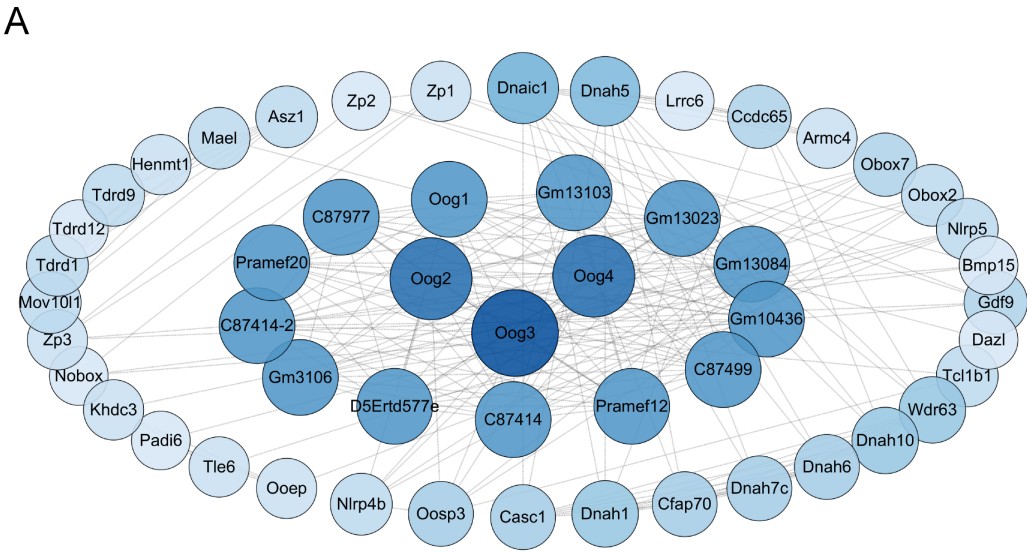

B

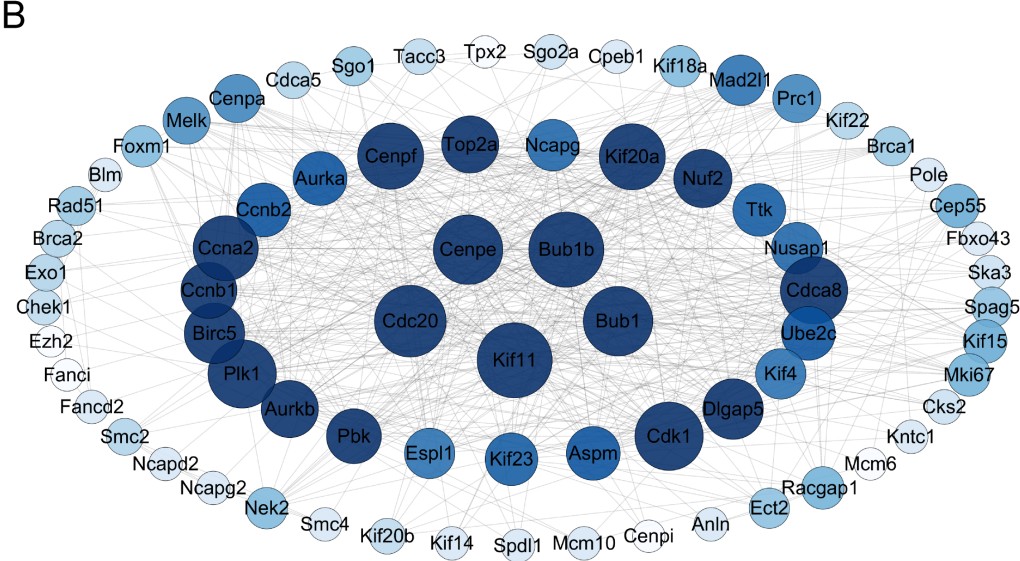

**Figure 5  Interaction maps of PPI networks between groups.** (A) Protein network interaction based on DEGs of V15 and CON. (B) Protein network interaction based on DEGs of V30 and CON. Colors from dark to light represent more to less node degrees, respectively. PPI, protein-protein interaction.

Changes in the serum hormone levels, follicle number, and estrous cycle in the mice indicated that a mouse model of POI was successfully established (Fig. 1). Analysis of differences revealed 467 DEGs in V15 and 2,349 DEGs in V30 compared to the controls group, this finding was corroborated by the difference seen in phenotypic characteristics. Functional enrichment analyses of downregulated DEGs and upregulated DEGs in V15 and V30 mice were then performed. The V15 group was significantly enriched only in the motor protein pathway. In addition to showing enrichment in the motor protein

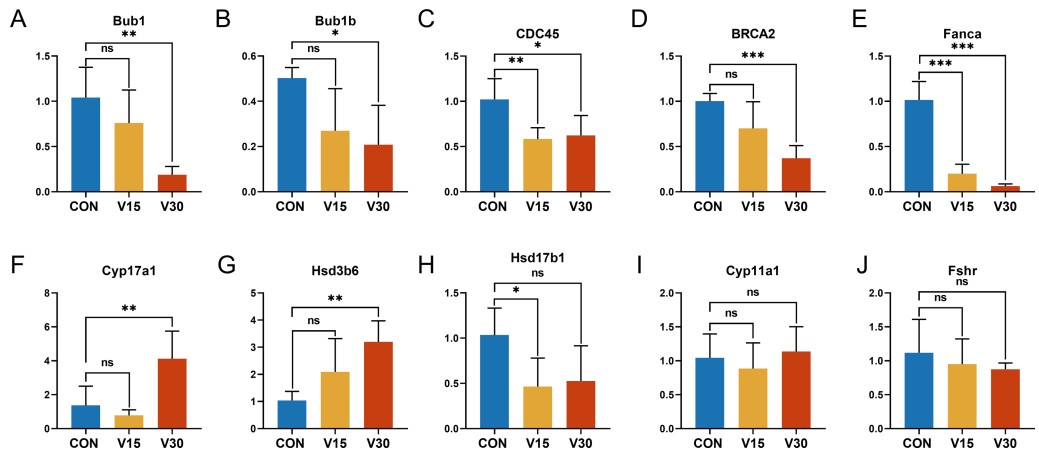

**Figure 6  Relative expression levels of target genes in the control mice and experimental mice.** Notes: *$p < 0.05$ compared to the control group; **$p < 0.01$ compared to control group; ***$p < 0.001$ compared to control group. The data are displayed as mean ± SD deviation ($n = 5$).

pathway, the functional enrichment analysis of V30 mice also indicated impaired ovarian steroid hormone synthesis and DNA damage repair (Figs. 4C and 4D). The differences in the number of DEGs in V15 and V30 mice and the enrichment results may suggest that VCD has a time-dependent effect of VCD on the ovary. Combined with the morphological features of mouse ovaries observed at different time points, these differences may be because VCD damages ovaries mainly through small follicles, affecting the normal recruitment of primordial follicles so that very few or no follicles can develop and mature normally (*Tran et al., 2018*). The reproductive toxicity of VCD specifically targets ovarian follicles, and those who have been exposed to VCD may experience follicular atresia without even realizing it, causing them to enter menopause prematurely without showing signs of menstrual cycle disruption (*Hoyer & Sipes, 2007*). By the time symptoms appear, the structure and function of the ovaries may already be extremely compromised.

Endocrine disruptors usually affect the activity of steroid-producing enzymes, leading to an imbalance of sex hormones. This finding is also consistent with the of the present study. CYP17A1 plays a critical role in the production of sex hormones and is associated with the development of many diseases (*Burris-Hiday & Scott, 2021*). Sex hormone production requires 17,20-lyase activity in CYP17A1, which acts in the early stages of the classical steroidogenic pathway. Severe mutations in CYP17A1 lead to 17-hydroxylase/17,20-lyase deficiency (17OHD), which disrupts steroidogenesis in both the adrenal glands and the gonads (*Auchus, 2017*). The expression level of CYP17A1 was upregulated in the present study, and it is hypothesized that this was due to VCD affecting the short negative feedback loop of estrogen-mediated steroidogenesis in the follicle (*Erickson et al., 1985*; *Taniguchi et al., 2007*). The 3 $\beta$-HSD enzymes exist in multiple isoforms; to date, six different isoforms have been identified in mice, namely, HSD3B1, HSD3B2, HSD3B3, HSD3B4, HSD3B5 and HSD3B6 (*Abbaszade et al., 1997*). As a key transcriptional regulator in the steroid hormone synthesis machinery, Hsd3b6 is involved in the biosynthesis and metabolism of a variety

of steroid hormones, such as progesterone, in females (*Falone et al., 2016*). Dysfunction of HSD3B6 expression can have deleterious effects on ovarian function and female fertility. According to previous studies of human and rodent granulosa cells, HSD7B1 plays a central role in $E_2$ biosynthesis (*Mindnich, Möller & Adamski, 2004*). One study knocked out the HSD17B1 gene in mice and reported impaired ovarian function, and a 10-fold reduction in HSD17B activity in the tissue homogenates of these mice. This finding suggests that other HSD17B proteins participating in the $E_1$ to $E_2$ or $E_2$ to $E_1$ transition have a much lower capacity to produce $E_2$ *in vivo* and are unable to compensate for the absence of HSD17B1 (*Ghersevich et al., 1994*; *Luu-The, Tremblay & Labrie, 2006*; *Hakkarainen et al., 2015*). Taken together, these findings indicate that VCD has an endocrine-disrupting effect, and direct induction of transcriptional changes in key genes may be one of the ways that VCD exerts this effect. VCD disrupts the endocrine system by disrupting the normal progression of steroid hormone synthesis and by affecting follicular growth and development.

The present study also revealed considerable deficiencies in DNA damage repair, specifically within the Fanconi anemia pathway, homologous recombination pathway, and oocyte mitosis pathway, as indicated by the GO and KEGG enrichment analyses. Previously studies have shown, *via* experimental methods such as whole exome sequencing (WES), that impaired DNA damage repair—damaged DNA molecular structure that cannot be reconstructed—is a leading cause of POI (*Jiao et al., 2018*). DNA double-strand breaks (DSB) are the most severe form of DNA damage. The main types of DSB repair are homologous recombination (HR) and nonhomologous end joining (NHEJ) (*Ensminger & Löbrich, 2020*; *Lingg, Rottenberg & Francica, 2022*). The Fanconi anemia (FA) pathway, a significant DNA damage repair mechanism, plays a pivotal role in this process (*Zhang et al., 2015*; *Gebel et al., 2020*). Recent studies have revealed multiple genes in the FA pathway that are involved in the development of primary ovarian insufficiency (POI). For instance, BRCA gene mutations increase the risk of related cancers, such as breast cancer, because of BRCA genes' relevance to the maintenance of genome integrity and, in particular, their important role in the homologous recombination (HR) DNA repair pathway (*Minello & Carreira, 2023*). At DSBs, BRCA1 plays a pivotal role in facilitating DNA-end resection (*Moynahan et al., 1999*; *Schlegel, Jodelka & Nunez, 2006*), while BRCA2 operates mainly downstream (*Schlacher, Wu & Jasin, 2012*). Maintenance of genome integrity requires both to act in concert against damage during replication (*Schlacher et al., 2011*; *Daza-Martin et al., 2019*). The enrichment analysis in the present study revealed significant reductions in several FA pathways and key genes involved in homologous recombination, including BRCA1, BRCA2, FANCA, FANCD2. Therefore, VCD may affect the efficiency of ovarian DNA repair by disrupting repair pathways such as the HR, which affects the structural integrity of DNA and ultimately leads to oocyte and granulosa cell apoptosis, follicular atresia, and POI.

Oocyte meiosis is an important aspect of the female reproductive process. Unlike in sperm maturation, female germ cells begin their first meiotic division during embryonic development and arrest at the bilobed stage of meiotic prophase. During estrus in animals or during the female menstrual cycle, which is stimulated by a rapid increase

in luteinizing hormone (LH) secreted by the pituitary gland, some of the stalled oocytes resume meiosis (*Mehlmann, 2005*). Unlike normal somatic cell division, meiosis in the oocytes is asymmetric, producing polar bodies and oocytes, with the polar bodies having very little cytoplasm and the oocytes having the vast majority of the cytoplasm. A previous study revealed that genetic variants in several genes involved in oocyte meiosis are associated with POI in humans (*Gordon, Kanaoka & Nelson, 2015*). However, many more genes have yet to be studied in relation to POI. Based on the data from the present study, three genes were chosen whose expression significantly differed: BUB1, BUB1B, and CDC45. BUB1 and BUB1B serve as spindle assembly checkpoints during oocyte meiosis and play crucial roles in spindle assembly checkpoint signaling as well as accurate chromosome alignment. Knocking down the BUB1 and BUB1B transcripts leads to a significant number of oocytes being blocked at the M1 stage, accompanied by a high percentage of abnormal spindle organization and chromosome misalignment (*Liu et al., 2010*). It has been demonstrated that silencing the BUB1B gene reduces the number of oocytes capable of expelling the first polar body to only 6% (*Homer, Gui & Carroll, 2009*). Reduced expression of the two entities could result in meiotic abnormalities in the oocyte. CDC45 is a pivotal checkpoint in the cell cycle that assumes an essential function in the eukaryotic DNA replication's inception and extension phases. By binding to with the MCM2-7 hexamer and the GINS complex (the CMG complex), this complex forms a replication deconjugating agent that triggers and initiates the process of DNA unwinding, but can hinder cellular replication if the replication deconjugating agent is faulty (*Moyer, Lewis & Botchan, 2006*). Cell division is regulated by checkpoints controlling both cytoplasmic and nuclear processes. Occurrences of abnormalities during cell division may result in the formation of defective cells, leading to cell death (*Barnum & O'Connell, 2014*). The effect of VCD on oocyte mitosis, including whether VCD over activated or blocked, deserves additional in-depth study.

This study included a limited number of mice, and the DEGs and enrichment pathways identified in the study need to be further validated in relevant female samples. In addition, more detailed animal experiments, such as those involving the isolation of oocytes and granulosa cells at different stages, should be designed in the future to ascertain the specific effects of VCD on different types and stages of cell populations.

## CONCLUSIONS

The reproductive toxicity of VCD is a significant potential threat to women who are exposed to VCD. The present study showed differences between day 15 and day 30 in damage caused by VCD to the primordial follicular pool. The results of this study suggest that ovarian steroid hormone synthesis, DNA damage repair, and oocyte mitotic mechanisms play key roles in POI and that the ovarian damage from VCD may be time dependent. The transcriptomic features presented in this study may provide guidance for POI population treatment strategies and may be useful for understanding the mechanism of oocyte mitosis.

## ACKNOWLEDGEMENTS

We are very grateful for the figure support that provided by Figdraw.

### Funding

This work was supported by the National Natural Science Foundation of China (No. 82360303, No. 81960278) and the Gansu Provincial Joint Research Foundation (No. 23JRRA1493). The funders had no role in study design, data collection and analysis, decision to publish, or preparation of the manuscript.

### Grant Disclosures

The following grant information was disclosed by the authors:
National Natural Science Foundation of China: 82360303, 81960278.
Gansu Provincial Joint Research Foundation: 23JRRA1493.

### Competing Interests

The authors declare there are no competing interests.

### Author Contributions

- Yi Li conceived and designed the experiments, performed the experiments, analyzed the data, prepared figures and/or tables, and approved the final draft.
- Ruifen He conceived and designed the experiments, performed the experiments, analyzed the data, authored or reviewed drafts of the article, and approved the final draft.
- Xue Qin performed the experiments, authored or reviewed drafts of the article, and approved the final draft.
- Qinying Zhu analyzed the data, authored or reviewed drafts of the article, and approved the final draft.
- Liangjian Ma analyzed the data, authored or reviewed drafts of the article, and approved the final draft.
- Xiaolei Liang conceived and designed the experiments, authored or reviewed drafts of the article, and approved the final draft.

### Animal Ethics

The following information was supplied relating to ethical approvals (*i.e.*, approving body and any reference numbers):

Laboratory Animal Center of Lanzhou University provided full approval for this research (LDYYLL2023-105).

### Data Availability

The data is available at NCBI GEO: GSE249151.

The raw measurements are available in the Supplemental Files.

### Supplemental Information

Supplemental information for this article can be found online at http://dx.doi.org/10.7717/peerj.17251#supplemental-information.

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
