# Peer review of "Transcriptome analysis during 4-vinylcyclohexene diepoxide exposure-induced premature ovarian insufficiency in mice"

_PeerJ, doi:10.7717/peerj.17251_

## Round 0.1 · original submission · Major Revisions

Please address the concerns of both reviewers and amend the manuscript appropriately.

Reviewer 1 ·

Basic reporting

In this manuscript, Li et al. made a mice POI model by intraperitoneal injection of VCD. Serum and ovaries were harvested from 2 different time points and compared to healthy mice from control group by a series of methods like sectioning, RNA sequencing, DEG, PPI network analysis, qRT-PCR, etc.
Overall, the result in this study was neat, and the discussion was informative. Over 2000 genes were reported to be either up- or down- regulated in the POI model, especially at 30-day time point, many of which could be related to motor protein pathway, ovarian steroid hormone synthesis and DNA damage repair. As a result, this manuscript would potentially provide guidance for future pathological and pharmaceutical studies.
The following are a few minor comments:
1. In Fig.1B, the change of hormone level looks to be quite low. Was this little change enough to lead to significant phenotype? Please explain.
2. V30 group showed more DEGs compared to V15 group. Could some of these DEGs in V30 group be a result of the DEGs in V15 group, i.e. downregulation of gene A after 15days lead to downregulation of gene B after 30 days?
3. It would be nice to have figure lagend for supplemental figure.
4. A few typos like in lane 42, delete one [However,]. In lane 74, [to access] should be [access to]. In lane 219, [V group] should be [V15 group].

Experimental design

The overall work flow applied in this study was standard. I have a few questions regarding the construction of POI model:
1. The concentration of VCD used was 160mg/kg. How was this concentration determined? If there was a reference, please specify.
2. It is a little unclear how exactly V30 group was treated. Were V30 treated with VCD for 30 days, or 15 days with VCD and another 15 days with sesame oil? Need clarification.

Validity of the findings

In the last part of this study, 10 genes were selected and their expression levels were compared across CTL, V15 and V30 samples by qRT-PCR. Before showing the qRT-PCR results, the authors need to give the reason for picking these 10 genes. Are they the most up/down regulated genes in the DEG analysis, or the most physiologically relevant genes?

Reviewer 2 ·

Basic reporting

The manuscript by Li et al. is aimed at developing a mouse model for VCD and understanding the pathogenesis mechanism that is responsible for POI. Overall, the study is successful in developing the model and identifying the key genes and proteins responsible for changing ovarian function using a combination of bioinformatic tools, RNA-sequencing, and RT-PCR. Particularly, the authors should be commended for appropriately describing the background and establishing the necessity of the study in the introduction, as well as the discussion section, where they explain the significance of their results by referencing previous research. However, there are several outstanding issues regarding the use of the English language and representation of figures. Moreover, certain conclusions drawn by the authors do not support the data, and I have detailed these issues in the subsequent parts of this review.

1. The clarity of the text can be enhanced through improvements in the English language for a global readership. I would suggest consulting a colleague well versed in English and informed regarding the subject matter. Alternatively, the authors can also consult a professional editing service.
2. The authors introduce acronyms, such as POI, in the abstract without providing the full form or acronym, which needs to be corrected.
3. The references in the introduction should be placed after each sentence and not clubbed together with multiple sentences.
4. In Fig. 1A, instead of Day 63, it should be Day 53.
5. A reference is needed for the statement on line 159.
6. The size of the graphs in Fig. 1C needs to be increased as the labels are unclear in their current form.
7. In Fig. 2A, the font size in the graph axes needs to be increased. In Fig. 2C &D, the labels overlap the data points and are unreadable. Maybe the authors can use arrows to label the points effectively. Fig. 2 legend has mentioned a Venn diagram, which is missing in the figure. Either the authors need to remove the text or add the figure. The authors need to add a concluding statement for the Transcriptomics-based identification of DEGs section describing the significance of the information obtained.
8. For Fig. 3 and 5, the font size of the text for log10 and count should be increased.
9. In Fig. 5, authors are advised to enlarge the figures, which will help them to increase the font size of the text in the nodes as well. Additionally, the authors can add a table of the PPI results in the supplementary with the node degrees in descending order i.e. highest node degree at the top and lowest at the bottom of the table.
10. The authors need to briefly (in one or two sentences) explain the significance of their findings at the end of the PPI Network Analyses of DEGs section.
11. The authors need to cite Fig. 6 at appropriate places for the Expression levels of target genes in the ovaries section. Additionally, each of the graphs in Fig. 6 needs to be denoted as A, B, C, and so on.
12. The authors need to explain in greater detail why they chose Bub1, Bub1b, Cdc45, Cyp11a1, Cyp17a1, 211 Hsd17b1, Hsd3b6, Fshr, Brca2, Fanca genes as it is unclear as to what analysis they are referring to when they mention ‘previously described analyses’ (line 210). Most of these genes were not explicitly mentioned to be important in any of the previous sections of the manuscript.

Experimental design

The study demonstrates a scientifically sound and largely well-executed experimental design. The materials and methods are clearly articulated, providing a comprehensible framework for potential replications. Nevertheless, I have outlined minor and major suggestions for your consideration below.

1. In line 109, the type of electron microscope needs to be mentioned.
2. More experimental details regarding each step in the construction of cDNA libraries need to be included in the methods section.

Validity of the findings

The conclusions do not always match the figures or outcomes of the experiments. A few instances have been listed below:

1. It is unclear why the authors suggest that C87414 and Pramef12 are among the top 5 key genes as they appear no different than, for example, Oog1 or C87414 based on their colors.
2. In lines 212-213, the authors state that the expression level of Bub1 and Bub1b decreases in the V15 group. However, based on Fig. 6, it seems that the slight decrease is not statistically significant.
3. The authors should remove Cyp11a1 and Fshr from the list of names on line 217, as there is no difference in the expression level between them.
4. The Cyp17a1, Hsd17b1, and Hsd3b6 graphs need statistical significance calculations for V15/V30 and the control group.
5. In lines 253-255, the authors refer to a previous study that shows the downregulation of Cyp17a1 and states it is similar to what they observed. However, that is not the case, as they observe an increase in gene expression in V30 compared to the control, and in V15, the expression is not affected. Similarly, in lines 258-261, the authors state copper nanoparticle exposure and VCD exposure show similar effects, which is untrue as VCD exposure increases Hsd3b6 expression rather than decreasing it.
6. The forward and reverse sequence for Bub1 primer needs to be verified as they are identical.

---

## Round 0.2 · Minor Revisions

Although both reviewers were mostly satisfied by the revision, one of them has pointed out: " since the line references ("line ##-##") in the response letter do not always agree with the revised manuscript, the authors are recommended to double check the revised manuscript and make sure the changes described in the response letter are actually included in the manuscript."

Please check the manuscript and make the necessary amendments.

Reviewer 1 ·

Basic reporting

All of my concerns were addressed in the revision.

Experimental design

My concerns were well clarified in the response letter.
However, since the line references ("line ##-##") in the response letter do not always agree with the revised manuscript, the authors are recommended to double check the revised manuscript and make sure the changes described in the response letter are actually included in the manuscript.
An example is the following statement does not agree with the revised manuscript: "The revised content is as follows: V15 and V30 were dosed daily with VCD (160 mg/kg) for 15 days. The control group was injected with sesame oil for 15days. The dose of VCD were based on published reports6 and our pre-experiments. Vaginal cytology was made daily during the modelling period to observe the estrous cycle. For V15 group, serum and ovaries were harvested on day 15 after modeling. For the CON and V30 group, serum and ovaries were collected on day 30 after modeling."

Validity of the findings

All of my concerns were addressed in the revision.

Reviewer 2 ·

Basic reporting

The authors have addressed the reviewer concerns and it should be accepted for publication.

Experimental design

No comment

Validity of the findings

No comment

---

## Round 0.3 · accepted · Accept

All final issues pointed out by the reviewer were adequately addressed and the revised manuscript is acceptable now.